# Protocol for a 2-year longitudinal study of eating disturbances, mental health problems and overuse injuries in rock climbers (CLIMB)

Klara Edlund [ID],[1,2] Isabel Nigicser [ID],[3] Mikael Sansone,[3] Fredrik Identeg,[4] Henrik Hedelin,[5] Niklas Forsberg,[6] Ulrika Tranaeus [ID] [7,8]

For numbered affiliations see end of article.

**Correspondence to**
Dr Klara Edlund;
klara.edlund@ki.se

## ABSTRACT

**Introduction** Rock climbing is a rapidly growing sport in which performance may be affected by participant's weight and leanness, and there may be pressure on athletes with respect to their eating behaviour and body weight. However, there is sparse research performed on climbers, constituting a knowledge gap which the present study aims to fill. The primary outcomes of the study are to examine disordered eating and overuse injuries in rock climbers. Secondary variables are body image, indicators of relative energy deficiency, mental health problems, compulsive training, perfectionism, sleep quality and bone density.

**Method and analysis** This prospective longitudinal study aims to recruit Swedish competitive rock climbers (>13 years) via the Swedish Climbing Federation. A non-athlete control group will be recruited via social media (n=equal of the climbing group). Data will be collected using streamlined validated web-based questionnaires with three follow-ups over 2 years. Inclusion criteria for rock climbers will be a minimum advanced level according to International Rock-Climbing Research Association. The non-athlete control group is matched for age and gender. Exclusion criteria are having competed at an elite level in any sport as well as training more often than twice per week. Statistical analyses will include multinominal logistic regression, multivariate analysis of variance (MANOVA) and structural equation modelling (SEM). We will assess effect measure modification when relevant and conduct sensitivity analyses to assess the impact of lost to follow-up.

**Ethics and dissemination** The Rock-Climbers' Longitudinal attitudes towards Injuries, Mental health and Body image study, CLIMB, was approved by the Swedish ethics authority (2021-05557-01). Results will be disseminated through peer-reviewed research papers, reports, research conferences, student theses and stakeholder communications.

**Trial registration number** NCT05587270.

## STRENGTHS AND LIMITATIONS OF THIS STUDY

⇒ Longitudinal assessment of disturbed eating, mental health status, overuse injuries, bone health and symptoms indicative of relative energy deficiency in advanced rock-rock climbers and normal controls.

⇒ The use of valid and reliable instruments and measurement methods and the large number of potential confounders will improve the internal validity of the estimated associations.

⇒ The use of dual-energy X-ray absorptiometry complements the participant's self-report data with objective measurements on bone health.

⇒ Multivariate statistical analysis will be executed to answer the research questions.

⇒ One limitation is the use of self-report questionnaires only rather than the combination of self-report and structured interviews.

## INTRODUCTION

Rock climbing as a sport places high demands on strength, mobility and endurance of the athlete. In sports, like rock climbing, weight and leanness may play an important role for achievement, and there may be pressure on athletes with respect to their eating behaviour and body weight.[1] In a study by Giel *et al*[2] on elite athletes in 51 Olympic sport disciplines, the results indicated higher rates of eating disorders (EDs) symptoms among athletes in weight dependent sports as well as higher levels of mental health problems. However, there is limited evidence and mixed results in the literature on ED pathology in elite athletes.

Rock climbing has in recent years gained increased popularity as a recreational and competitive sport, as reflected by a rapid increase of practitioners, media coverage and the recent inclusion of the sport in the Olympics of 2020 and 2024. Research on climbing related disordered eating (DE) and relative energy deficiency in sport (RED-S) is sparse. The recent increase in the numbers of rock climbers, both competing athletes and recreational rock climbers, has brought attention to health issues related to climbing. Especially for high-performance climbing, relative

strength to weight ratio has been shown to be of importance for performance resulting in lean body shapes, commonly with low Body Mass Index (BMI), fat percentages and low calorie intakes.[3]

DE is characterised by a deep dissatisfaction with one's own body and/or shape, and is associated with life-threatening medical and psychiatric comorbidities.[4] The prevalence of clinical and subclinical EDs in elite athletes has been reported to be 13%–16% higher than for the general population.[5–7] In sports where the importance of low body weight is considerable, athletes who compete appear to be particularly vulnerable to the development of disturbed eating patterns.[8] This includes sports such as ballet, gymnastics and figure skating.[9] Even at the non-professional level, performers of lean sports emphasising thinness and muscularity such as bodybuilding and ballet exhibit DE behaviours.[10] Suggested sport-specific, risk factors for eating disturbances include frequent weight regulation, dieting and experienced pressure to lose weight.[11][12] The International Federation of Sport Climbing (IFSC) has recently implemented a lower weight limit of BMI>17.5 $kg/m^2$ for eligibility to compete in international competition since low body weight (BMI<17.5 $kg/m^2$) has repeatedly been observed among competitors. To the same end, the Austrian Sport Climbing Organisation has set a lower age limit for competitors: 18 years for males and 17 years for females to mitigate the risk of the developing eating disturbances.[13]

RED-S is a syndrome referring to impaired physiological functioning caused by relative energy deficiency and includes, but is not limited to, impairments of metabolic rate, menstrual function, bone health, immunity, protein synthesis and cardiovascular health. Low energy availability (LEA) appears to be an important variable, as studies report that LEA impairs reproductive function and bone formation. As well as decreasing bone formation, it has also been seen to induce a higher increase bone resorption in women compared with men.[14] A recent study by Joubert et al[15] examined 114 female rock climbers competing at the World Cup level and found that 15.8% presented with current amenorrhoea. Additionally, among these rock climbers with amenorrhoea, a larger per cent revealed having struggled with DE compared with those without menstrual disturbances (13.5% vs 22.5%, respectively). Studies with a more comprehensive examination of the aetiology of RED-S, performance and injuries among rock climbers have been called for.

Compulsive exercise (CE) consists of maladaptive compensatory behaviours and serves as a strategy for emotion regulation—often closely linked to eating disturbances.[16] Weight-affecting behaviours in non-clinical groups have demonstrated addictive qualities that aid the understanding of why these behaviours are difficult to change.[16] Weight-affecting behaviours such as CE have a marked negative effect on psychological and physiological health,[17] and emotion dysregulation has been suggested to trigger these behaviours.[18] A previous study of university students indicated a negative association between

symptoms of depression and CE for weight control, and a positive association for body dissatisfaction and depression.[19] These results are in line with studies suggesting excessive exercise to be associated with different psychological traits where poor emotion regulation and compulsivity are two of them.[20] This may suggest that CE serves as a negatively reinforced behaviour, serving as short-term alleviation of discomfort. Furthermore, exercising for regulation of negative emotion has been consistently identified as a contributory factor to the development and maintenance of EDs.[21] This contribution does not lie specifically in the presence of negative emotion, but rather in the CE habits that develop as an outlet for these emotions, which in turn is a risk factor for developing disturbed eating. CE is characterised by an inability or unwillingness to cut down or stop the behaviour despite adverse health consequences.

Body dissatisfaction is associated with a drive for thinness,[22][23] dieting[24] and DE patterns.[24][25] The exposure to physical ideals has been seen to have a stronger immediate impact on girls than boys,[26] and preoccupation with weight, maladaptive methods for weight control and body dissatisfaction are known precursors of more serious clinical EDs, both among athletes and in the general population.[11][12][27] However, athletes from leanness-focused sports report higher rates of eating disturbances compared with athletes from non-weight sensitive sports.[9]

Empirical research suggests that elite athletes involved in lean-sensitive or weight-sensitive sports are at elevated risk of developing an ED.[1][28] In 2020, one of the first papers on EDs and climbing ability was published, which assessed 604 rock climbers (ages 32±9 years). It presented demographic data according to the International Rock-Climbing Research Association's (IRCRA) directives.[29] The results showed that higher level rock climbers had significantly lower BMI among both genders, and that 8.6% of the rock climbers reported DE behaviours. Additionally, 4.2% reported having been treated for an ED and there was an increased risk for EDs among female elite rock climbers compared with males.[30] Strand[31] states in a study of conversations in climbing communities that conversations related to achieving or maintaining low weight and lean body shape for performance enhancement purposes are common. Furthermore, Strand advises clinicians to be attentive to 'weight talk' in the climbing community.

Our main hypothesis is that elite and advanced rock climbers will present more DE symptoms compared with controls, with elite rock climbers showing more symptoms than advanced rock climbers. A secondary hypothesis is that rock climbers who report more eating disturbances will present with higher levels of health problems including decreased bone density, CE, depression, body dissatisfaction, menstrual disturbances and more overuse injuries as compared with rock climbers with less symptoms of eating disturbances, as these secondary variables are correlates of EDs.[32] We expect these patterns to be stable over time. The authors will also examine the gender

differences in these categories and expect, for example, to find higher percentages of DE among women than men, following the pattern of the presented literature. At the time of writing, there exists sparse research on the connection between rock climbing, DE and energy deficiency. The presented hypotheses are therefore largely grounded on findings in other sports with similar bodily and/or weight requirements, such as ballet, figure skating and gymnastics. Due to the rapidly increasing popularity of recreational and competitive climbing, longitudinal cohort studies are, therefore, urgently needed to understand the aetiology of health problems related to EDs in rock climbers in particular. Studies that identify risk and prognostic factors are required to develop health promotion and disease prevention strategies, which ultimately can reduce the burden of disease in rock climbers worldwide.

## METHOD AND ANALYSIS
### Study design and participants
CLIMB will be executed as a prospective longitudinal study, using web-based questionnaires at four occasions (baseline, 6 months, 1, 2, 3 and 5 years follow-up). The follow-up frequency is established considering that rock climbing as a sport does not have strict seasons and is highly individualistic, necessitating more frequent measurements. The whole eligible population of Swedish rock climbers on elite or advanced levels according to the IRCRA will be invited to participate in the study. Information about the study will be distributed by the Swedish Climbing Federation (https://www.klatterforbundet.se) via climbing gyms and national teams. Information about the study will also be promoted on social media channels such as Facebook and Instagram. Participants will be both male and female, 13 years and older.

The inclusion criteria for the climbing group follow the grade classifications of IRCRA.[29] Each climber must climb at a minimum of advanced level. For women, this means having climbed a boulder route rated harder than 6A or a lead route rated 6c within the last year. For men, this means having climbed a boulder route rated harder than 6B+ or a lead route rated 7a+ within the last year.

A non-athlete control group (n=equal of the climbing group) from the general population, matched in age and gender, will be invited to participate in the study. They will be recruited via social media and universities in Gothenburg and Stockholm, and they will be recruited for cross-sectional analysis only and will not be invited to participate in the follow-ups. The control group's eligibility will be established by answering a two-question survey regarding general physical and sedentary activity (detailed in the Instruments section), as well as being above 13 years of age. Exclusion factors for a control participant would be to have trained and competed in any sport at the elite level prior to answering the questionnaire, as week as training more than twice per week. One cannot be too inactive to participate in the control group.

### Patient and public involvement
None.

### Instruments
The first portion of the questionnaire aims to confirm that participants are eligible for inclusion, with a climbing level of advanced or above as classified by IRCRA. Demographic data will be gathered and reported including factors such as age, time practising the sport, amount of weekly training hours, climbing level and gender. Symptoms related to eating disturbances, RED-S, body image, compulsive training, perfectionism, mental health problems, sleep quality, overuse injuries and menstrual dysfunction will thereafter be measured (see table 1).

**Table 1** Measurements in the CLIMB-study.

| Variable | Instrument | Baseline | Follow-up (6 months, 1, 2, 3 and 5 years postbaseline) |
|---|---|---|---|
| Eating disorder symptoms | The Eating Disorders Questionnaire (Fairburn and Beglin, 1994)[33] | X | X |
| Overuse injuries | The Oslo Sports Trauma Research Centre Overuse Injury Questionnaire (Clarsen et al, 2013)[35] | X | X |
| Perfectionism | The Frost Multidimensional Perfectionism Scale (Frost et al)[36] | X | X |
| Compulsive training | The Compulsive Exercise Test (Taranis et al)[39] | X | X |
| Body dissatisfaction | Body Shape Questionnaire, brief version (Welch et al)[41] | X | X |
| Mental health | Depression Anxiety Stress Scale-21 (Lovibond and Lovibond)[44] | X | X |
| Sleep quality | The Pittsburg Sleep Quality Index (Carpenter and Andrykowski)[46] | X | X |
| Symptoms of relative energy deficiency | Relative Energy Deficiency in Sports: RED-S (Edlund Health Q, Edlund, 2020)[50] | X | X |
| Physical activity and sedentary behaviour | Indicator Questions from the Swedish National Board of Health and Welfare | X* | |
| Bone density | DEXA | X† | |

*Physical activity and sedentary behaviour will only be performed on the control group.
†DEXA will only be performed on a proportion of rock climbers.
DEXA, dual-energy X-ray absorptiometry.

In addition to the web-based questionnaire, bone health using dual-energy X-ray absorptiometry (DEXA) will be measured for climbing participants at baseline.

### Eating Disorders Examination Questionnaire

DE will be measured by the Eating Disorders Examination Questionnaire (EDE-Q 6.0),[33] which is validated and includes 36 questions and records symptoms during the 28 days prior to filling in the questionnaire (eg, 'Have you been deliberately trying to limit the amount of food you eat to influence your shape or weight (whether or not you have succeeded)?'. EDE-Q consists of four subscales (Restriction Scale, Eating Concern, Figure Concern and Weight Concern) as well as a score indicating global level of EDs symptoms. A cut-off score of >2.5 will be used as indicative of DE and >4.0 for EDs. EDE-Q has demonstrated adequate test–retest reliability (r=0.89) and internal consistency (α=0.85).[34]

### The Oslo Sports Trauma Research Centre Overuse Injury Questionnaire

The Oslo Sports Trauma Research Centre Overuse Injury Questionnaire (OSTRC-O)[35] is used to gauge the effects of pain and injuries on participation in sport. The original questionnaire measures injuries in the knee, shoulder, foot/lower leg and lower back. Since climbing is a sport relying heavily on chest, neck, finger, hand and arm strength, additional items targeting these body parts will be added to the instrument. The questionnaire consists of four questions regarding the athlete's participation in training and competition, reduced training, performance and pain the asked period (eg, 'have you had any difficulties participating in normal training and competition due to (physical) problems last week?' The alternatives for responses are ranged from 'full participation without problems', 'full participation, but with problems', 'reduced participation due to problems' to 'cannot participate due to problems'. The responses are scored from 0 to 25 for each question with 0 representing no problem and 25 maximal level of the problem. The total score for the OSTRC-0 is 100. The questionnaire shows high internal consistency, with a Cronbach's α of 0.91.[35]

### Perfectionism

Perfectionism is measured by the Frost Multidimensional Perfectionism Scale (FMPS).[36] It consists of 35 questions, which are measured on a scale from 1 to 5 depicting the applicability of each statement to the participant. The questions are divided to give scores in six subcategories, namely: concern over mistake, personal standards, parental expectations, parental criticism, doubts about action and organisation (eg, 'It is important to me that I be thoroughly competent in what I do'). A cut-off score of >29 points is used in clinical trials.[37] Two subscales will be assessed in the present study due to these subscales showing acceptable to good internal consistency (personal standards=0.74; concern over mistakes=0.86).[38]

### Compulsive Exercise Test

CE is measured by the Compulsive Exercise Test (CET),[39] which is composed of 24 items with 5 subscales: avoidance and rule driven behaviour, weight control exercise, mood improvement, lack of exercise enjoyment, and exercise rigidity (eg, 'I exercise to burn calories and lose weight'). For CLIMB, only the subscales avoidance and rule-driven behaviour and weight control exercise will be included in the analysis since they are deemed the most relevant in athletes and have been reported to have adequate validity (factor analysis) and internal consistency (weight control exercise: a=0.82, avoidance and rule-driven behaviour: a=0.87).[40]

### The Body Shape Questionnaire short version

The Body Shape Questionnaire short version (BSQ-8C)[41] is used to measure body dissatisfaction among participants (eg, 'Have you felt so bad about your shape that you have cried?'). This short 8-question version is derived from the original 34-item BSQ, and it is shown to be reliable and valid.[42] We use the Swedish BSQ-8C which has been found to exhibit high internal consistency (α=0.94) as well as excellent test–retest properties.[43]

### Depression Anxiety and Stress Scale

Mental health problems are measured by the Depression Anxiety and Stress Scale.[44] The 21 items consider the week prior to answering the survey and observe symptoms related to depression, anxiety and stress (eg, depression: 'I felt that I had nothing to look forward to'; anxiety: 'I was aware of dryness of my mouth'; stress: 'I found it difficult to relax'). The cut-off scores group participants into groups showing normal, mild, moderate, severe or extremely severe symptoms. The exact cut-off values differ in each of the three categories. It is the most widely used version of the survey and boasts test–retest and an internal consistency of α=0.81–0.96.[44 45]

### Pittsburgh Sleep Quality Index

Sleep quality is measured by the Pittsburgh Sleep Quality Index (PSQI),[46] which includes 19 items and evaluates 7 subcategories: Subjective sleep quality, sleep latency, sleep duration, habitual sleep efficiency, sleep disturbances, use of sleeping medications and daytime dysfunction (eg, 'During the past month, how often have you had trouble sleeping because you… wake up in the middle of the night or early morning') . A Global Score is composed of these categories on a scale of 0–21, where '0' indicates no sleep difficulty and '21' indicates severe difficulties in all areas. The PSQI has shown adequate internal consistency, reliability and construct validity. Cronbach's alphas have been reported to 0.80 across groups and correlations between global and component scores have been shown to be moderate to high.[46]

Symptoms of RED-S will be measured by Edlund Health Q, a 15-item Swedish survey focusing on menstrual cycles, injury and eating habits.

Physical activity and sedentary behaviour are measured by two indicator questions established by the Swedish National Board of Health and Welfare (Socialstyrelsen).[47] The first item is regarding the amount of high intensity activity measured in time per week, such as cardiovascular training. The second item asks the amount of low intensity activity measured in time per week, such as walking or gardening. These two questions are only answered by the control group to establish eligibility among the participants.

DEXA will be used for measuring bone mineral density.[48] This non-invasive imaging technology uses a small amount of ionising radiation to quantify the amount of bone, fat and lean tissue, granting the examiners the opportunity to assess the risk of fracture as well as form an overall perception of the bone's objective health and nutritional status.

The web-based questionnaire first introduces the purpose, aims and ethics of CLIMB, emphasising the anonymity of the participant. If the participant is <15 years of age, consent from both guardians is required. The questionnaire will be answered by all participants at baseline (rock climbers and controls), but the follow-up questionnaires will only be completed by the climbing group. This is to facilitate future data collection and analysis, as the control group data will only be analysed at baseline as part of a cross-sectional study. The follow-up measurements are, therefore, only required from the rock climbers in order to provide longitudinal patterns. DEXA will only be performed on rock climbers at baseline.

## Statistical methods
### Sample size
No previous data on injury incidence based on overuse injuries exist for rock climbers. Based on previous retrospective data from rock climbers,[49] using a power of 80%, a significance level of 5% and an expected relative risk of 1.5 for the primary outcome, we calculated that approximately 55 participants were needed in each group (rock climbers and control group).

### Main exposure variables
DE behaviour, body image, overuse injuries/injuries, indicators of LEA, mental health problems, compulsive training, perfectionism, sleep quality, sedentary behaviour and bone density.

### Outcome
The outcome ED will be measured with the EDE-Q. The EDE-Q consist of four subscales, restrained eating, eating concern, shape concern and weight concern addressing core dimensions of EDs. A score of >4 is indicative of an ED, and for the present study a score of >2.5 will be used as an indication of subclinical eating disturbances. Thus, higher scores are indicative of more severe symptoms of ED pathology.

Our primary outcomes are EDs symptoms measured with EDE-Q and overuse injuries measures with OSTRC-O.

Our secondary outcomes are bone density (DEXA), symptoms of RED-S (Edlund Health Q), mental health problems (Depression, Anxiety and Stress Scale), sleep quality (PSQI), perfectionism (FMPS), CET and body dissatisfaction (BSQ-8).

### Data analysis plan
Multinominal logistic regression analyses will be used to determine the associations between each of the exposures and trajectories of outcomes. The associations as ORs and 95% CIs will be reported. Bivariate models to measure the crude associations between the exposures and trajectories will be built.

The differences between the rock climbers and controls regarding the above-mentioned outcome measure ED will be analysed using multivariate analysis of variance (MANOVA). Both within-group and between-group analysis will be conducted. Further analyses among rock climbers with high ED versus low ED will be compared in relation to bone density, injuries, mental health and symptoms of RED-S. The MANOVA will be used for that purpose. A one-way repeated measures ANOVA will be used to identify changes over the 2-year span in the two groups of rock climbers with high ED versus low ED. In cases where overall significant changes are detected, post hoc analysis will be applied to specify where the between-group differences exist.

Structural equation modelling will be used at a later stage to investigate if the variables body image, injuries, mental illness, sports environment, bone density (from baseline) and EDs can predict injuries.

## ETHICS AND DISSEMINATION
All participants will provide informed consent to participate in CLIMB after they had been informed about the purpose and procedure of the study and that it has been approved by the Swedish Ethical Review Authority (reference number: 2021-05557-01).

The dissemination plan is to present our results to the climbing and sport communities, in peer-reviewed scientific journals, at congresses and to stakeholders with influence in the development of climbing environments.

### Study status
The manuscript reports the protocol (ClinicalTrials.gov NCT05587270) for an ongoing study for which participants are currently ongoing. Data collection has been initiated and baseline analyses are planned to commence during 2023.

## DISCUSSION
Climbing is a weight sensitive sport where athletes generally benefit from a lean body shape and low weight[6]—a risk factor for eating disturbances. There are very few studies regarding eating disturbances among rock climbers. A study by Joubert et al, examining DE among international

sport lead rock climbers, displayed that 6.3% of male and 16.5% of female climbers had DE.[30] Similarly, a study examining amenorrhoea among female competitive climbers showed that 15.8% of female climbers had amenorrhoea. The IFSC recently released a statement to counteract this development of the sport. Due to the over-representation of eating disturbances reported in other lean sports and the poor treatment prognosis, early detection and prevention of eating disturbances and energy deficiency is essential, especially considering the growing popularity of the sport.

There are several major strengths of CLIMB. The first is the use of longitudinal data within several domains related to disturbed eating and mental health issues by using validated questionnaires, reducing misclassification of outcomes. The few studies published at this time, on this topic, do not consider multiple mental health parameters that CLIMB intends to address, such as perfectionism, sleep habits, body dissatisfaction and CE. It is the first study on rock climbers which also includes a control group, providing vital context on disease prevalence. CLIMB will be representative of the advanced climbing community at large in Sweden as it includes a wide age span of both genders who climb at advanced and elite levels around Sweden, markedly decreasing the risk of selection bias. Additionally, this large sample of climbers contributes to ensuring external validity. The use of DEXA complements the participant's self-report data with objective measurements on the effects that relative energy deficiency is associated with, establishing a point of action and initiative to prevent and treat afflicted athletes.

One limitation of our study is that we are using a non-random convenience sample that may not be representative to advanced rock-rock climbers in an international perspective. We have aimed to include the whole advanced level rock-climbing community in Sweden aged 15 years and older, which is the rationale for how the participants were recruited. According to the Swedish Climbing Association, there were 15 409 climbers affiliated to a climbing club in 2022. It is unknown, however, how many climbers are of an advanced or higher level, and thus eligible for participating in the study. Another potential limitation is the use of self-report questionnaires only which yield subjective data. Although the authors will encourage all participants to fill out all surveys using communication strategies at baseline and throughout the follow-up surveys, there is an undeniable risk of increased drop-out rates with each passing survey.

The climbing-concerned scientific community is at the beginning of forming a paradigm around this subject. The many facets of this study make CLIMB invaluable in gaining a holistic view on each participant as well as on group levels, which ultimately contributes to developing and implementing evidence-based interventions around the globe.

**Author affiliations**

¹Department of Health Promotion Science, Sophiahemmet University, Stockholm, Sweden

²Unit of Intervention Research on Worker Health, Institute of Environmental Medicine, Karolinska Institute, Stockholm, Sweden

³Ortopedi, Sahlgrenska Academy, Goteborg, Sweden

⁴Sahlgrenska Academy, Goteborg, Sweden

⁵Sahlgrenska University Hospital, Goteborg, Sweden

⁶N/A, Stockholm, Sweden

⁷Department of FNB, Swedish School of Sport and Health Sciences, Stockholm, Sweden

⁸Institute of Environmental Medicine, Karolinska Institute, Stockholm, Sweden

**Contributors** MS is the project director and KE the co-project director, and they have together with the senior researchers HH and UT contributed to the planning of the study as well as the conception and design. They, together with PhD students IN and FI have contributed to writing this study protocol. NF is a research assistant in the project. They have contributed to the planning, conception and design of the study including planning and executing the digital survey. They have all contributed to the recruitment of the participants and administered the work around setting up the study. UT has also performed the DEXA measures. All authors have contributed to the writing process with input, feedback and critically revising this study protocol. The manuscript was approved by all authors.

**Funding** The authors have not declared a specific grant for this research from any funding agency in the public, commercial or not-for-profit sectors.

**Competing interests** None declared.

**Patient and public involvement** Patients and/or the public were involved in the design, or conduct, or reporting, or dissemination plans of this research. Refer to the Methods section for further details.

**Patient consent for publication** Not applicable.

**Provenance and peer review** Not commissioned; externally peer reviewed.

**ORCID iDs**

Klara Edlund http://orcid.org/0000-0002-2614-5174

Isabel Nigicser http://orcid.org/0000-0002-0707-7276

Ulrika Tranaeus http://orcid.org/0000-0002-2102-6352

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
