## [Reviewer comments · BMJ Open]

ARTICLE DETAILS

TITLE (PROVISIONAL)	Protocol for a Two-Year Longitudinal Study of Eating Disturbances, Mental Health Problems and Overuse Injuries in Rock-Climbers (CLIMB)
AUTHORS	Edlund, Klara; Nigicser, Isabel; Sansone, Mikael; Identeg, Fredrik; Hedelin, Henrik; Forsberg, Niklas; Tranaeus, Ulrika

VERSION 1 – REVIEW

REVIEWER	Lubans, David The University of Newcastle, School of Education
REVIEW RETURNED	03-May-2023

GENERAL COMMENTS	General Comments Competitive climbing is a weight-sensitive sport where performance advantages are seen where athletes have a low bodyweight relative to their strength. This environment may encourage behaviours detrimental to one's physical and mental health, particularly amongst elite and sub-elite populations. This study aims to measure the body composition, disturbed eating (DE), mental health status, overuse injuries, bone health, and symptoms indicative of relative energy deficiency (RED-S) in competitive climbers in Sweden as well as in a control population. I commend the authors on identifying a gap in the research and proposing this study. Below, I have made many specific recommendations intended to improve this particular protocol, as well as the study itself. Specific Comments Abstract Final dot point - I'm not sure that the use of self-report questionnaires would increase the risk of attrition, or that this is worth mentioning in the abstract. Introduction Throughout the paper, please specify that you are referring to 'rock climbing'. Currently, the opening paragraph is difficult to follow and could do with some restructuring to help to focus it. You can start the paragraph with the sentence beginning on line 9 "Climbing as a..."; mention that such a focus on weight may contribute to the development of disordered eating; provide some evidence on disordered eating in elite sports/sports with a focus on weight; then
--

highlight the gap in this area of research. Mention of amenorrhea in climbers doesn't seem to belong in this opening paragraph. It would work better in your third paragraph of the introduction.

Please provide evidence of your point that exercising for the regulation of negative emotion contributes to the development of eating disorders. The reference you supplied does not provide evidence of this.

Second last paragraph in intro - This should not be a standalone paragraph. Please integrate into other paragraph/s. Did the study you cite mention whether the prevalence of eating disorders was higher in their sample than the general population? Also, you should speak more specifically to the current body of evidence. Mentioning single studies here and there can be misleading, as the broader literature may say something different. If you're mentioning a single study because there is very little else out there, you should make this clearer. Also, you have mentioned this study to further the case that the link between weight and performance in climbing may contribute to eating disorders, but it's rather disjointed at the moment. 1 - if it's there, please synthesise the available evidence that weight/performance formula contributes to eating disorders and poorer mental health, or 2 - if that explicit link between these potential predictors and outcomes isn't clear, make more of a case as to what piece is missing and how this study may address it.

Final paragraph in introduction - no need for the first sentence. Regarding the title, you aren't actually measuring climbers' attitudes towards these outcomes (mental health, injury, etc.), you're measuring the constructs themselves (i.e. you're measuring their actual mental health, not their attitude towards their mental health). I know this seems picky, but we're in the business of being nit-picky! You should also adjust your title accordingly.

Final paragraph in introduction - I see your footnote about the interchangeability of 'sub-elite' and 'advanced' climbers. I think the paper would benefit from a brief indication of how you determine what an 'elite' or 'sub-elite' climber is provided in-text.

Regarding your second hypothesis, please provide citations of the relationship between eating disturbances and the other outcomes you mention.

Methods

Study design... - better phrasing would be "using web-based questionnaires measured at four occasions: baseline...etc.,)"

Study design... - you position the 'additional follow-ups' as distinct from the other four measurement occasions. Are you measuring the same outcomes? If so, you can just mention these timepoints along with the others.

Study design... - You mention that you will invite the population of Swedish elite or sub-elite (rock) climbers to the study; do you have some indication of how many people this is? You do not mention a number here but propose to recruit 180 people as a control group. Where has this number come from?

	You should have more explicit inclusion/exclusion criteria. For your control group, will you include people who participate in other weight-focused sports (including those is sub-elite rock climbing)? For both groups, will you exclude people with disabilities, those who are clinically depressed, anxious, or who already have eating disorders? Further, you mention that eligibility in the control group will be established by a two-question physical activity survey. What are the actual exclusion criteria though? Are participants excluded if they are too inactive or too active? Please be precise with this. Regarding the DEXA measurements - what proportion of climbers will be measured and what is your plan for this data? Given that you're using DEXA, are you reporting on specific body composition metrics? In the methods, you should mention that the control group is being recruited for a cross-sectional analysis that you have planned, and that they will not complete follow-up measures. Instruments (line 31) - remove 'poor' from 'poor sleep quality' Table 1 - please revise to include each measurement timepoint as columns (baseline, 6 months, 12 months, etc.,) Measures - PA and Sedentary Behaviour - Measuring PA and sedentary behaviour in both groups will allow you to include it as a covariate in your analyses. Measures - For each of your measures, please provide evidence of their validity, and provide an example item from the scale. Measures - Overuse Questionnaire - please include a more explicit example of the items you intend to add. Methods - you need to indicate what modifications you are making to your survey for your follow-up. You should also specify why you have so many follow-up timepoints. Currently, your research question and hypotheses do not necessitate the number of follow-ups, nor the time between them. For such a long span of time, you could look at predictors of drop-out from the sport. You can measure changes in their mental health either throughout a season (pre competition, during, post season), or look at your outcomes in participants before and after they dropout of the sport. Alternatively, if your primary interest is the relationship between these variables in competitive rock-climbers, you could answer this question in a much shorter span of time (say over 12 months). I think you need to formulate your hypotheses to align with your proposed timeframe and number of measures. Statistical methods - Power calculation - you should be more specific here. What outcome was the power calculation based on? When you refer to a difference between groups, what is the difference between groups that your study is powered to detect? Also, when you mention group differences, you are referring to the cross-sectional analysis mentioned earlier in the paper. What is the sample size required for your longitudinal study?
--	--

	Data analysis plan - SEM analysis - using longitudinal data, you have an opportunity to test whether certain qualities of the climbers are predictive of your outcomes of interest. You may want to think about proposing some other models. You could look at whether compulsive training predicts DE or vice-versa. You could look at the temporality of body dissatisfaction and depression/anxiety (which is more predictive of the other?). I encourage the authors to consider more questions they may be able to answer with this data. Discussion Discussion - Please move your second paragraph to a separate limitations section. Otherwise, move it to penultimate paragraph of the discussion section. Generally, the discussion section needs reworking. Please use the discussion to succinctly reiterate the problem and focus on the importance of this study, with particular attention paid to specific research questions that it may be able to answer. The discussion of osteoporosis and anorexia seem only peripherally related to this study and don't belong in the discussion of this protocol.
--	---

REVIEWER	Burtscher, Martin University of Innsbruck, Sport Science, Medical Section
REVIEW RETURNED	18-May-2023

GENERAL COMMENTS	The authors address an important research question from a scientific and practical/clinical perspective as well. Generally, some more available data on the frequency of risks to be evaluated in climbers may be provided in order to underline the importance of this study. You state that “climbing has rapidly grown in popularity”; do you have numbers? Can you provide an approximate number of Swedish competitive climbers? This will be necessary and important for the assessment of the validity of your study. Will the vast majority of climbers be willing to participate in order to avoid a severe selection bias? At least 40 participants will be included in each group, but it remains unclear how many groups you will have (based on age, sex, skill level, etc.). Who of the authors will take biostatistical responsibility? The “instruments” used seem sound to me. When do you expect the start of the study? Do you already have a provisional financial plan? In the discussion section you may a bit more elaborate on efforts made (in the past and recently) in the “climbing world” to counteract the risk of eating disorders, etc. and also discuss study findings that have already dealt with these problems.
---

VERSION 1 – AUTHOR RESPONSE

Reviewer #1

Prof. David Lubans, The University of Newcastle

General comments:

Competitive climbing is a weight-sensitive sport where performance advantages are seen where athletes have a low bodyweight relative to their strength. This environment may encourage behaviours detrimental to one's physical and mental health, particularly amongst elite and sub-elite populations. This study aims to measure the body composition, disturbed eating (DE), mental health status, overuse injuries, bone health, and symptoms indicative of relative energy deficiency (RED-S) in competitive climbers in Sweden as well as in a control population. I commend the authors on identifying a gap in the research and proposing this study. Below, I have made many specific recommendations intended to improve this particular protocol, as well as the study itself.

Authors' response:

Thank you for your feedback on our proposed study protocol and longitudinal study of advanced rock-climbers (and controls) in Sweden.

Specific comments:

Abstract:

Final dot point - I'm not sure that the use of self-report questionnaires would increase the risk of attrition, or that this is worth mentioning in the abstract

Authors' response:

We have talked this point over and agree with you and the editor and have edited this dot point and a paragraph in the discussion section accordingly.

Introduction:

1. Throughout the paper, please specify that you are referring to 'rock climbing'.

Authors' response:

We have changed the wording from "climbers" to "rock-climbers" through the paper.

2. Currently, the opening paragraph is difficult to follow and could do with some restructuring to help to focus it. You can start the paragraph with the sentence beginning on line 9 "Climbing as a..."; mention that such a focus on weight may contribute to the development of disordered eating; provide some evidence on disordered eating in elite sports/sports with a focus on weight; then highlight the gap in this area of research. Mention of amenorrhea in climbers doesn't seem to belong in this opening paragraph. It would work better in your third paragraph of the introduction.

Authors' response:

The opening paragraph and the portion regarding amenorrhea have been edited as suggested.

3. Please provide evidence of your point that exercising for the regulation of negative emotion contributes to the development of eating disorders. The reference you supplied does not provide evidence of this.

Authors' response:

The paragraph about compulsive exercise has been updated and re-written as suggested.

4. Second last paragraph in intro - This should not be a standalone paragraph. Please integrate into other paragraph/s. Did the study you cite mention whether the prevalence of eating disorders was higher in their sample than the general population? Also, you should speak more specifically to the current body of evidence. Mentioning single studies here and there can be misleading, as the broader literature may say something different. If you're mentioning a single study because there is very little else out there, you should make this clearer. Also, you have mentioned this study to further the case that the link between weight and performance in climbing may contribute to eating disorders, but it's rather disjointed at the moment. 1 - if it's there, please synthesise the available evidence that weight/performance formula contributes to eating disorders and poorer mental health, or 2 - if that explicit link between these potential predictors and outcomes isn't clear, make more of a case as to what piece is missing and how this study may address it.

Authors' response:

The second last paragraph has been updated based on the suggestions given.

5. Final paragraph in introduction - no need for the first sentence. Regarding the title, you aren't actually measuring climbers' attitudes towards these outcomes (mental health, injury, etc.), you're measuring the constructs themselves (i.e. you're measuring their actual mental health, not their attitude towards their mental health). I know this seems picky, but we're in the business of being nit-picky! You should also adjust your title accordingly.

Authors' response:

The first sentence in the final paragraph has been deleted as suggested.

Yes, we're in the business of nit-picky and we appreciate the thoroughness of the comments. We have changed the title as suggested.

6. Final paragraph in introduction - I see your footnote about the interchangeability of 'sub-elite' and 'advanced' climbers. I think the paper would benefit from a brief indication of how you determine what an 'elite' or 'sub-elite' climber is provided in-text.

Authors' response:

We have changed "sub-elite" to "advanced" throughout the paper.

7. Regarding your second hypothesis, please provide citations of the relationship between eating disturbances and the other outcomes you mention.

Author's response:

We have added a reference to the IOC consensus statement (Mountjoy et al, 2018) where RED-S and the relationship between eating disturbances and our other variables are discussed.

Methods

1. Study design... - better phrasing would be "using web-based questionnaires measured at four occasions: baseline...etc.,)"

Authors' response:

The wording has been edited as suggested.

2. Study design... - you position the 'additional follow-ups' as distinct from the other four measurement occasions. Are you measuring the same outcomes? If so, you can just mention these timepoints along with the others.

Authors' response:

This phrase has been edited as suggested.

3. Study design... – You mention that you will invite the population of Swedish elite or sub-elite (rock) climbers to the study; do you have some indication of how many people this is? You do not mention a number here but propose to recruit 180 people as a control group. Where has this number come from?

Authors' response:

We have added information regarding the number of participants affiliated to a climbing club in 2022 in the discussion. It is not however known, which level these climbers are at, and it is thus unknown how many of these are eligible for the study. We have changed the $n=180$ to $n=\text{equal of the control group}$. 180 was an estimate of how many climbers we could possibly include.

After consulting with the Swedish climbing association and based on the number of climbers participating in competition climbing at elite or sub-elite level in Sweden the number 180 was chosen to represent a substantial part of the Swedish climbers at that level. There is no formal registration of climbers in Sweden so the actual number of climbers at any given level is by definition an estimate. This is also stated in the discussion section.

4. You should have more explicit inclusion/exclusion criteria. For your control group, will you include people who participate in other weight-focused sports (including those in sub-elite rock climbing)? For both groups, will you exclude people with disabilities, those who are clinically depressed, anxious, or who already have eating disorders? Further, you mention that eligibility in the control group will be established by a two-question physical activity survey. What are the actual exclusion criteria though? Are participants excluded if they are too inactive or too active? Please be precise with this.

Authors' response:

We have clarified according to the suggestion in the last paragraph of the "study design and participants." The outcome measures of the study include observing the prevalence of depression, anxiety, and eating disorders, and are therefore not exclusion criteria for either group.

5. Regarding the DEXA measurements - what proportion of climbers will be measured and what is your plan for this data? Given that you're using DEXA, are you reporting on specific body composition metrics?

Authors' response:

Sweden is a large country with considerable distances between towns, and DEXA-measurements are difficult to carry out on multiple places due to logistic considerations. We invite all climbing participants to participate in the DEXA-measurements. Due to the amount of traveling required to participate in these for some participants, we chose not to make this mandatory, to avoid limiting the sample of possible participants. We hope to include at least 30% of all climbing participants in the DEXA-measurements. Specific body composition measurements, including BMI, FMI, FFMI and BMD will be reported.

6. In the methods, you should mention that the control group is being recruited for a cross-sectional analysis that you have planned, and that they will not complete follow-up measures.

Authors' response:

This paragraph has been updated as suggested.

7. Instruments (line 31) - remove 'poor' from 'poor sleep quality'

Authors' response:

"Poor" has been removed as suggested.

8. Table 1 - please revise to include each measurement timepoint as columns (baseline, 6 months, 12 months, etc.,)

Authors' response:

This has been clarified in Table 1.

9. Measures - PA and Sedentary Behaviour - Measuring PA and sedentary behaviour in both groups will allow you to include it as a covariate in your analyses.

Authors' response:

Since the control group did not consist of only athletes, we chose to use the most used, validated questionnaire in Sweden for non-athletes regarding the subject. The questionnaire is developed by the National Board of Social Affairs and Health and unfortunately only measures physical activity.

10. Measures - For each of your measures, please provide evidence of their validity, and provide an example item from the scale.

Authors' response:

This has been updated as suggested.

11. Measures - Overuse Questionnaire - please include a more explicit example of the items you intend to add.

Authors' response:

This has been updated as suggested.

12. Methods - you need to indicate what modifications you are making to your survey for your follow-up.

Authors' response:

Thank you for highlighting this. The team has discussed and have decided to keep the survey unchanged for the follow ups, assuming there is no discomfort reported by the participants. Should this be the case, certain items may be removed if they are deemed to not change over a period of time rendering them unnecessary to be measured at all follow-up points. The sentence reading "A slightly shorter version of the survey will be used for the follow up measures..." has therefore been removed.

13. You should also specify why you have so many follow-up timepoints. Currently, your research question and hypotheses do not necessitate the number of follow-ups, nor the time between them. For such a long span of time, you could look at predictors of drop-out from the sport. You can measure changes in their mental health either throughout a season (pre competition, during, post season), or look at your outcomes in participants before and after they drop-out of the sport.

Authors' response:

This is a good point and the planned number of follow-up-timepoints may be overly ambitious. Based on earlier research experience the research group does, however, aim for redundancy. This redundancy can both, to a certain degree mitigate the effects of loss of data from any given time point and also give a more comprehensive bulk of data.

Rock climbing, as a sport, does not have strict seasons and practitioners commonly compete in multiple overlapping sub-disciplines. The drop-out concept is also not comparable to e.g gymnastics or many other sports since elite climbers mostly train alone or with comrades rather than always with a trainer or coach. This individualistic and non-structured aspect of climbing is probably one of the reasons why the questions we are pursuing in this study plan have not been explored before to any greater extent. A sentence regarding this has been added to the first paragraph of "Method and Analysis."

14. Alternatively, if your primary interest is the relationship between these variables in competitive rock-climbers, you could answer this question in a much shorter span of time (say over 12 months). I think you need to formulate your hypotheses to align with your proposed timeframe and number of measures.

Authors' response:

We appreciate this perspective. We would, however, like to emphasize that most of these factors are unknown in climbing and we believe a 12- month time frame would be too short even for this. For reasons outlined under point 13 we have opted for rather frequent follow-ups.

15. Statistical methods - Power calculation - you should be more specific here. What outcome was the power calculation based on? When you refer to a difference between groups, what is the difference between groups that your study is powered to detect? Also, when you mention group differences, you are referring to the cross-sectional analysis mentioned earlier in the paper. What is the sample size required for your longitudinal study?

Authors' response:

The power analysis has been updated and are now based on previous research on injuries in rock climbing.

16. Data analysis plan - SEM analysis - using longitudinal data, you have an opportunity to test whether certain qualities of the climbers are predictive of your outcomes of interest. You may want to think about proposing some other models. You could look at whether compulsive training predicts DE or vice-versa. You could look at the temporality of body dissatisfaction and depression/anxiety (which is more predictive of the other?). I encourage the authors to consider more questions they may be able to answer with this data.

Authors' response:

Thank you for your suggestions. It is detailed in the Data analysis plan that between-group differences will be measured longitudinally with ANOVA. In significant cases, post-hoc analysis will be applied for

further specification. Temporality of body dissatisfaction and depression/anxiety will be noted and potentially analyzed in this manner.

Discussion

1. Discussion - Please move your second paragraph to a separate limitations section. Otherwise, move it to penultimate paragraph of the discussion section.

Authors' response:

The updated paragraph has been moved as suggested.

2. Generally, the discussion section needs reworking. Please use the discussion to succinctly reiterate the problem and focus on the importance of this study, with particular attention paid to specific research questions that it may be able to answer. The discussion of osteoporosis and anorexia seem only peripherally related to this study and don't belong in the discussion of this protocol.

Authors response:

Thank you for your feedback. We have made the discussion more succinct and reworked the entire section.

Reviewer #2:

Dr. Martin Burtcher, University of Innsbruck

Reviewer's comments:

1. The authors address an important research question from a scientific and practical/clinical perspective as well.

Author's response:

Thank you. Yes we do want to emphasize the need for practical/clinical perspectives in parallel to our research data.

2. Generally, some more available data on the frequency of risks to be evaluated in climbers may be provided in order to underline the importance of this study.

Authors' response:

Absolutely. At the moment of writing, however, there exists very sparse research regarding the connection between rock-climbing and eating disorders/energy deficiency. Our hypotheses are therefore largely based on findings from other sports with similar bodily/weight requirements. The importance of this study lies therefore in establishing a foundation in this new field. We have added two sentences regarding this in the last paragraph of the introduction and a smaller section in the discussion regarding the existing literature among climbers.

3. You state that "climbing has rapidly grown in popularity"; do you have numbers?

Can you provide an approximate number of Swedish competitive climbers? This will be necessary and important for the assessment of the validity of your study.

Will the vast majority of climbers be willing to participate in order to avoid a severe selection bias?

Authors' response:

We have added information regarding the number of participants affiliated to a climbing club in 2022 in the discussion. It is not however known, which level these climbers are at, and it is thus unknown how many of these are eligible for the study.

4. At least 40 participants will be included in each group, but it remains unclear how many groups you will have (based on age, sex, skill level, etc.).

Authors' response:

We have clarified which groups are referred to in the methods-section.

5. Who of the authors will take biostatistical responsibility?

Authors' response:

Authors Klara Edlund and Mikael Sansone will take biostatistical responsibility.

6. The "instruments" used seem sound to me.

Authors' response:

Yes, we believe that the instruments chosen have firm psychometric properties.

7. When do you expect the start of the study?

Authors' response:

At this time the data collection has begun.

8. Do you already have a provisional financial plan?

Authors' response:

There is a financial plan for this study. A sentence stating this has been added to the "trial status" paragraph.

9. In the discussion section you may a bit more elaborate on efforts made (in the past and recently) in the "climbing world" to counteract the risk of eating disorders, etc. and also discuss study findings that have already dealt with these problems.

Authors' response:

Thank you for your comment. We have added a section regarding previous studies on the subject among climbers and similarly recent measures to counteract the development, published by the International federation of sport climbing.

VERSION 2 – REVIEW

REVIEWER	Lubans, David The University of Newcastle, School of Education
REVIEW RETURNED	11-Aug-2023
GENERAL COMMENTS	Dr Levi Wade also contributed to the review of this manuscript.

REVIEWER	Burtscher, Martin University of Innsbruck, Sport Science, Medical Section
REVIEW RETURNED	13-Aug-2023

GENERAL COMMENTS	The authors responded adequately to my comments and I feel that this project could be carried out successfully. Best wishes!
---

VERSION 2 – AUTHOR RESPONSE